# Development of a conceptual model for evaluating new non-curative and curative therapies for sickle cell disease

**Kate M. Johnson**[1,2,3], **Boshen Jiao**[3], **M. A. Bender**[4], **Scott D. Ramsey**[3,5], **Beth Devine**[3], **Anirban Basu**[3,6]*

**1** Faculty of Pharmaceutical Sciences, Collaboration for Outcomes Research and Evaluation (CORE), University of British Columbia, Vancouver, Canada, **2** Faculty of Medicine, Division of Respiratory Medicine, University of British Columbia, Vancouver, Canada, **3** Department of Pharmacy, The Comparative Health Outcomes, Policy & Economics (CHOICE) Institute, University of Washington, Seattle, Washington, United States of America, **4** Clinical Research Division, Department of Pediatrics, University of Washington, Fred Hutchinson Cancer Research Center, Seattle, Washington, United States of America, **5** Division of Public Health Sciences and Hutchinson Institute for Cancer Outcomes Research, Fred Hutchinson Cancer Research Center, Seattle, Washington, United States of America, **6** Department of Health Systems and Population Health and Department of Economics, University of Washington, Seattle, Washington, United States of America

* basua@uw.edu

**Funding:** This research was funded in part by the National Institutes of Health (NIH) Agreements OT3HL152448 and OT3HL151434. The views and conclusions contained in this document are those

## Abstract

### Background

Sickle cell disease (SCD) is a clinically heterogeneous disease with many acute and chronic complications driven by ongoing vaso-occlusion and hemolysis. It causes a disproportionate burden on Black and Hispanic communities. Our objective was to follow the SMDM/ISPOR Task Force recommendations for good practices and create a conceptual model of the progression of SCD under current clinical practice to inform cost-effectiveness analyses (CEA) of promising curative therapies in the pipeline over a lifetime horizon.

### Methods

We used consultations with experts, providers, and patients to identify acute events and chronic conditions in the conceptual model. We compared our model structure to previous CEA models of interventions for SCD, assessed the prevalence of the identified disease attributes in Medicaid and Medicare claims databases, and identified relevant outcomes following the 2nd Panel in CEA. We determined an appropriate modeling technique and relevant data sources for parameterizing the model.

### Results

The conceptual model structure included four dimensions of disease: chronic pain, acute events, chronic conditions, and treatment complications, spanning 26 disease attributes with significant impacts on health-related quality of life and resource. We modeled chronic pain separately to reflect its importance to patients and interaction with all other disease attributes. We identified additional data sources for health state utilities and non-medical costs and benefits of SCD. We will use a microsimulation model with age- and sex-specific

of the authors and should not be interpreted as representing the official policies, either expressed or implied, of the NIH. The funders had no role in study design, data collection and analysis, decision to publish, or preparation of the manuscript.

**Competing interests:** The authors have no competing interests.

transitions between health states predicted by patient demographic characteristics and disease history.

## Conclusion

Developing the model structure through an explicit process of model conceptualization can increase the transparency and accuracy of results. We will populate the conceptual model with the data sources described and evaluate the cost-effectiveness of curative therapies.

## 1. Introduction

Sickle cell disease (SCD) is a group of inherited blood disorders that affects over 20 million people globally and approximately 100,000 people in the U.S [1]. It is characterized by the deformation of red blood cells into a crescent shape and is associated with recurrent vaso-occlusion and hemolysis that contribute to acute episodes of pain, tissue ischemia, inflammation, and progressive organ damage [2]. The consequences of SCD are multi-system, with a wide range of acute and chronic complications that afflict patients throughout their lifetimes and reduce life expectancy, quality of life, and the productivity of both patients and their caregivers [3–6].

A few non-curative therapeutics like L-glutamine, Crizanlizumab, and Voxelotor have been introduced recently. There is also hope for curative therapies on the horizon. Curative therapies are time-limited interventions that offer durable clinical benefits by correcting or attenuating the underlying condition [7]. The only proven curative treatment currently available for SCD is hematopoietic stem cell transplant (HSCT), which results in disease-free survival in >90% of younger patients with a human leucocyte antigen-identical sibling donors [8]. However, suitable donors are rare, and complications such as graft failure and graft-vs-host disease are common [9].

Genetic therapies that alter cellular DNA, increase non-sickling hemoglobin, or modify the expression of genes have the potential to provide the same benefits as HSCT without some of the complications associated with an allogenic source of stem cells [9]. A genetic treatment for beta-thalassemia, a related blood disorder, was recently approved in Europe [10]. With numerous other therapies currently undergoing clinical testing, the arrival of a gene-based therapy for SCD in the U.S. market is imminent [7,11]. The potential for a curative therapy that is accessible to the broader SCD population offers crucial hope for patients, but numerous barriers to uptake remain. There are concerns of an increased risk of malignancies, the durability of clinical benefits, and affordability with traditional funding mechanisms [12–15].

Decision-analytic modeling provides a framework for assessing the value of curative therapies and exploring uncertainties in long-term health outcomes. However, assessing the cost-effectiveness of emerging therapies, curative and non-curative, for SCD presents unique challenges. Evidence of clinical effectiveness will likely be based on trials with few patients, short timeframes, and surrogate endpoints. Most studies do not consider evidence on the effects of treatment outside the healthcare sector and on non-patient populations such as caregivers, even though these are significant issues for SCD patients and families [16,17]. Previous cost-effectiveness models of SCD have assessed interventions over short time horizons that did not capture the full trajectory of disease burden and treatment complications over a patient's lifetime [18–20]. Decisions regarding the exclusion of potentially relevant disease complications were not justified through an explicit process of model conceptualization.

To ensure the structure of the cost-effectiveness model is an appropriate representation of the problem of interest, a systematic and transparent process of model conceptualization should be undertaken. The conceptual model outlines the disease states and events that should be included and how these components interact with each other [21]. Model conceptualization is typically conducted as a series of consensus-building exercises in which disease experts consider alternative representations of the model system [22–24]. Despite strong evidence that different formulations of the model structure can lead to vastly different results, model conceptualization is infrequently reported and often driven by data availability rather than a contextual understanding of the disease [25–27]. This is a critical barrier to the credibility of model results. Similar to publishing a study protocol, decisions regarding the exclusion of certain disease attributes, the type of evidence chosen to inform the model, and the modeling technique should be determined through an explicit process and justified in the context of the decision problem [21,28].

Following best practice guidelines on model conceptualization and conceptual models developed in other disease areas [23,24,28,29], the objective of this paper is to outline the decision problem and describe the development of a conceptual model for the progression of SCD under current treatment practices, including the occurrence of pain episodes, acute events, chronic conditions, and treatment complications. This Model for Economic Analysis of Sickle Cell Cure (MEASURE) is being developed as part of the National Heart Lung and Blood Institute Cure Sickle Cell Initiative (https://curesickle.org) and will be used to evaluate curative and non-curative therapies for SCD.

## 2. Methods

We followed the Society for Medical Decision Making (SMDM)/ International Society for Pharmacoeconomics and Outcomes Research (ISPOR) Modelling Task Force recommendations for model conceptualization [28]. This process has two distinct components. First, we defined the decision problem and objectives of the model. Second, we used expert consultation, a systematic literature review of previous cost-effectiveness models of SCD, the Second Panel recommendations for cost-effectiveness analysis, and empirical analyses to determine which disease attributes and outcomes to include in the model structure [30]. The resulting conceptual model directed our choice of modeling technique and data sources for parameterizing the model.

### 2.1 Conceptualizing the problem

The Cure Sickle Cell Initiative is a national consortium aiming to accelerate gene-based curative therapies for SCD. Within this Initiative, the Clinical and Economic Impact Analysis (CEIA) team was tasked with developing simulation models to demonstrate the potential national impact of specific curative therapies for SCD and the distribution of that impact on patients and their families, payers, and employers. Our team is comprised of health economists, health services researchers and evidence synthesis specialists, clinicians, and economic modelers. An expert panel consisting of pediatric and adult-focused clinicians who care for patients with SCD, healthcare payers, patients, patient advocates, and ethicists support our activities through quarterly meetings.

The objectives and scope of the decision model are summarized in **Table 1**. We aim to provide a flexible framework for addressing questions on the value of curative therapies to various stakeholders. These include: (1) the cost-effectiveness of new non-curative therapeutics, genetic therapies, and HSCT versus standard therapies, (2) the comparative effectiveness of specific gene therapies in the developmental pipeline, and (3) the distribution of value based

**Table 1. Summary of model objectives.**

| Decision problem | Evaluate genetic therapies and HSCT for SCD |
|---|---|
| Funding source | National Heart Lung and Blood Institute Cure Sickle Cell Initiative |
| Disease | Sickle cell disease |
| Perspective | Societal |
| Target population | Individuals with SCD in the U.S. |
| Health outcomes | Rates of comorbidities and treatment complications, life years and quality-adjusted life years gained |
| Strategies/ comparators | Curative therapies compared to Common care (treatment with hydroxyurea and transfusion) |
| Resources/costs | Healthcare utilization, related and unrelated medical costs, time-use costs, productivity loss, unpaid caregiver-time costs |
| Time horizon | Lifetime |

HSCT: Hematopoietic stem cell transplant; SCD: Sickle cell disease.

on patient ages and disease severities at the time of intervention. A societal perspective is most appropriate for representing these multiple aspects of value, which is also recommended by best practice guidelines [30]. However, because of the treatment cost implications for insurers, we also will present our results from the public and private health care sector perspectives.

Our target population is individuals with SCD in the U.S. who are eligible for these novel therapies, as well as a control group to serve as a comparator population. We will evaluate the effects of curative therapies over a lifetime horizon to account for the potential for highly durable clinical benefits. (A claims-based definition of the target population is included in Section 2.5). We will consider clinical and economic outcomes, including comorbidities and treatment complications, life years, and quality-adjusted life years (QALYs) gained under comparator strategies. The associated resources are healthcare utilization, total and out-of-pocket medical costs, future medical costs unrelated to SCD, and non-medical costs such as time use costs, paid and unpaid productivity loss for patients and caregivers. An impact inventory is provided in the S1 Appendix, Table A2 in S1 Appendix [30].

## 2.2. Conceptualizing the model

To develop a conceptual model, disease attributes that are within the scope of the decision problem and strongly associated with the outcomes of interest should be identified. We consulted with disease experts, providers, and patients to establish a candidate set of acute events and chronic disorders that have a high impact on our outcomes of interest and the potential to be modified by curative therapies. We compared our conceptual model to previous cost-effectiveness models of SCD using a systematic review of the literature and analyzed the prevalence of disease attributes in Medicaid and Medicare claims databases [31].

## 2.3 Expert consultation

As advocated by modeling guidelines, we used focus groups and discussion meetings to solicit the opinions of clinical experts and patient representatives in model conceptualization [28,32,33]. We convened several groups for deliberations. The first group was internal to the CEIA team and consisted of a clinician who cares for patients with SCD (M.B.), an internist (S.R.), two health economists (A.B. and J.R.), and one health services researcher (B.D.). The team met regularly over a three-month period in late 2019 to identify a candidate list of disease attributes in each of the following categories: acute events, chronic disorders, and complications

associated with common procedures and treatments for SCD. Deliberations reflected the need to include the full range of potentially relevant conditions, with future discussion groups and empirical analyses intended to reduce the list of disease attributes to those most significant to the decision problem.

We presented the candidate list of disease attributes developed by the CEIA team to approximately 45 members of the CureSCi consortium at an in-person meeting held in November 2019. The meeting consisted of patients and patient advocates, clinicians, investigators, and representatives from the funder. We conducted roundtable discussions with groups of approximately five members who were asked to prioritize the candidate disease attributes or identify additional attributes based on their impact on model outcomes. Members of the CEIA team recorded additional disease attributes identified or recategorization of existing attributes. In December 2019, we conducted a second discussion session with our standing multi-disciplinary panel of SCD experts to assess the expanded list of disease attributes. Based on these two rounds of expert consultation, we produced a finalized list of disease attributes for the conceptual model. The expert panel assessed face validity of a diagrammatic representation of the model structure.

### 2.4 Literature review

We conducted a systematic review of published cost-effectiveness models of SCD. The methods are detailed in a separate publication [31]. Briefly, we searched PubMed, Embase, National Health System Economic Evaluation Database, the Tufts University CEA Registry, and EconLit for cost-effectiveness analyses published in English between 2008 and 2021, and white papers published between 2018 and 2021. Studies were eligible if they assessed any intervention, treatment complication, or screening program targeted to patients with SCD. We extracted data on the disease attributes and treatment complications included in the model structure and the modeling technique employed.

### 2.5 Empirical analyses of disease attributes

Because SCD is a condition where patients experience a multitude of related and unrelated complications, we used empirical analyses to prioritize the long list of disease attributes identified by our stakeholders. Specifically, we based this prioritization on the prevalence of these disease attributes among individuals diagnosed with SCD in public and private payer insurance claims databases, so that accurate risk prediction models can be developed over patient lifetimes. These databases are detailed in Section 2.6. In the public payer databases, we constructed separate cohorts of Medicare, Medicaid, and dual-eligible individuals meeting a validated case definition of one inpatient or two outpatient or emergency department claims for SCD (International Classification of Disease [ICD] 9th edition 282.6, 282.41, 282.42, ICD 10th edition D57, D57.8, excluding sickle cell trait 282.5 and D57.3) in any position during the study period (2008–2016) [34]. Individuals were followed from their index date; the first instance of utilization with a diagnosis of SCD in any position during enrollment, as this was seen as an opportunity to receive curative therapy. Demographic characteristics (age, sex, race, region) were ascertained at index, and individuals were required to be continuously enrolled for 12 months following the index date. All outcomes and utilizations were obtained on an annual basis. Individuals were followed until death, the end of enrollment, or the year preceding HSCT. Among patients with multiple periods of enrollment during the study period, only the longest enrollment period was used. A cohort schematic is shown in the Figure A1 in S1 Appendix.

We developed a list of ICD-9 and ICD-10 codes for each of the candidate disease attributes and assessed their prevalence using inpatient codes in any position or any E.D. or ambulatory

service codes (Table A3 in S1 Appendix). Chronic disorders were divided into two categories, subacute and chronic. Because subacute disorders have the potential for remission, we required a disease-related claim in any setting or position within the previous two years for the disorder to be considered present in the current year. Chronic conditions were present for all observations following the initial diagnosis. We assessed the prevalence of all disease attributes over a patient lifetime in the Medicaid, Medicare, and dual-eligibility cohorts. Disease attributes with extremely low prevalence across all three datasets were eligible for exclusion from the model structure. The prevalence of disease attributes in the Truven MarketScan database was calculated using the same methodology and is the subject of a separate publication.

## 2.6 Data sources

**2.6.1 Public and private insurance claims.**   Prior to model conceptualization, we identified two primary data sources for assessing the prevalence of disease attributes and populating the simulation model. Truven MarketScan is a national claims database of over 115 million beneficiaries from all 50 states with employer-sponsored private health insurance. These data include demographic information, outpatient and inpatient medical claims, prescription drug claims, and healthcare utilization records of enrollees. The Centers for Medicaid and Medicare Services (CMS) Medicaid Analytic eXtract database and Medicare Part A and B Fee-for-Service claims includes enrollment information and administrative claims for 100% of individuals eligible for Medicaid, Medicare, or with dual-eligibility in nearly all 50 states. These databases will be used to develop prediction equations for transition probabilities, rates of acute events, treatment patterns, treatment complications, and healthcare utilization.

**2.6.2 Health-related quality of life.**   Through model conceptualization, we identified several additional data sources needed to populate the decision model and assess the cost-utility of curative therapies. The Sickle Cell Clinical Research and Intervention Program (SCCRIP) is a prospective cohort study that enrolled patients diagnosed with SCD at five institutions in the U.S. and assessed their health outcomes over lifetime follow up [35]. Information on patient demographics and disease history was determined from medical chart review. HrQoL is collected through the PedsQL [36], a disease-specific instrument for children and adolescents. Established algorithms will be used to map scores on the PedsQL to health state utilities (HSUs) in order to calculate QALYs [37]. We will use this data to generate prediction indices for HSUs in pediatric and adult patients based on their demographic characteristics and health status in the current year. Although extrapolating pediatric data to the entire population is a limitation, a systematic review revealed no empirically estimated HSUs for patients with SCD in the US [38]. Studies from other regions provided utilities for a narrow set of complications, which were often estimated in non-SCD populations and did not account for patient age. Experts indicated that individual level data with empirically estimated HSUs was necessary to account for the complex interactions between disease attributes. We will assess the validity of our prediction indices in the adult population using published estimates of HSUs for adults with SCD from other regions [38].

**2.6.3 Non-medical costs.**   We will measure the incremental costs of labor market production, household production, volunteer activities, care for the household and non-household members, and time seeking medical care for patients with SCD. These costs will be derived by multiplying time-use spent on each of the above categories by the mean wage plus fringe benefits reported by the U.S. Bureau of Labor Statistics [39]. We will calculate time-use based on its association with HrQoL, which was recently established using data from the nationally representative American Time-Use Survey with a Well-being Module that measured self-perceived quality of life on a visual analog scale (subsequently converted to HrQoL) [40]. We will use

this method to calculate time-use costs based on predicted HSUs of patients with SCD. Future unrelated healthcare costs and end-of-life costs will be incorporated using age-specific reference estimates [41]. We will incorporate estimates from the literature of unpaid caregiver-time costs [6]. Medical costs and health state utilities of caregivers have not been reported in the literature or in other available data sources [38,42].

## 3. Results

We identified 13 acute events and 13 chronic disorders (26 conditions) to include in the model structure. These disease attributes reflect the complex and multi-system impacts of SCD over a patient's lifetime and the potential for curative therapies to have wide-ranging downstream consequences. The initial round of expert consultation resulted in five acute events and two chronic disorders added to the candidate list. However, two chronic disorders (problems with healthy growth and severe anemia) were removed due to a lack of appropriate diagnostic codes for identifying them in claims databases. Seizure disorder was included in the candidate list generated by the CEIA team but later removed as it is more commonly associated with complications of HSCT than standard therapies such as hydroxyurea or transfusions [43]. We presented the refined list of disease attributes to the expert panel, who included one additional acute event (dactylitis). Chronic pain, leg ulcers, asthma, sleep-disordered breathing, hepatobiliary complications, and liver disease, chronic mental health disorders, and fatigue were classified as subacute disorders. The complete set of disease attributes is shown in **Fig 1**.

### 3.1 Literature review

We identified 11 cost-effectiveness models of interventions or screening programs for SCD. Details of these models have been published elsewhere [31]. Here we provide a summary of those results. Only five models included any acute events or chronic disorders in the model structure. The remaining six models were trial-based, life-table models, or only considered

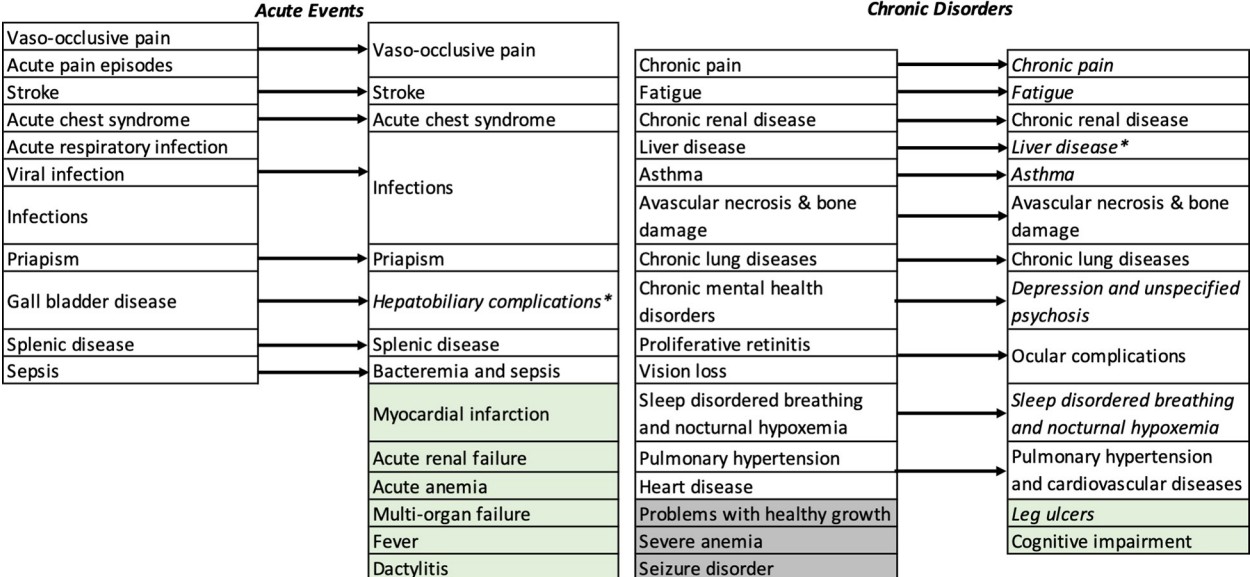

**Fig 1. Disease attributes considered for inclusion in the model structure.** Arrows indicate the refined categorization following expert consultation. Attributes shown in green were added and attributes shown in grey were removed. Italicized disease attributes were classified as subacute disorders. * Combined following empirical analysis of prevalence in MarketScan commercial claims database and classified as a subacute disorder.

treatment complications [18,20,44–47]. Stroke and vaso-occlusive events were both included in the structure of four models [19,48–51], acute chest syndrome was included in three models [48–50], and splenic sequestration was included in two models [19,50]. Bradt et al. developed the only model with additional disease attributes [49]. No previous model was developed through a formal model conceptualization process. If alternative model structures were considered, they were not described.

The most common modeling technique was cohort-based Markov modeling, which was used in six out of eleven studies. One study employed agent-based simulation [19] and the remaining studies were trial-based or life-table models [18,44,47,50]. Although four studies used a lifetime horizon, no model captured the full complexity of disease and treatment over the lifetime of a patient or considered the impacts of SCD from a societal perspective in the base case analysis.

A key feature of this review is to identify an appropriate comparator that can serve as the control group for the evaluation of the newer non-curative and curative therapies. CEA models in the literature often use hydroxyurea as a comparator [31]. However, our review of the CMS databases and feedback from our expert panel suggested that many patients who would be eligible for these newer therapies would not be on hydroxyurea. Consequently, we decided to form our control group consisting of patients who are either receiving no treatment, or hydroxyurea, or transfusions (but did not receive HSCT). We call this modality 'Common Care' for individuals with SCD. We use the latest available data from CMS, which consists of the population of individuals with SCD who are publicly insured and ends before the introduction of any of the newer non-curative therapeutics, making them ideal to inform the control care arm of the evaluations.

## 3.2 Empirical analyses of disease attributes

The prevalence of acute events and chronic disorders are shown in **Table 2**. In the Medicaid cohort (N = 39,366, mean age at enrollment 16.9 [SD 13.9] years, 53.4% female), Medicare cohort (N = 6,522, mean age 62.3 [SD 16.2] years, 60.2% female), and dual-eligible cohort (N = 36,846, mean age 33.3 [SD 19.5] years, 58.1% female), infections were the most prevalent acute or subacute event over patient lifetimes (79.4%, 73.0%, and 82.1%, respectively). Among chronic disorders, cardiovascular disease, including pulmonary hypertension, had the highest prevalence in the Medicaid (35.5%), Medicare (71.1%), and dual-eligible (59.6%) cohorts. No disease attribute had a prevalence of <2% in the combined cohort. The prevalence of disease attributes was similar in MarketScan.

## 3.3. Modeling technique

Our choice of modeling technique stemmed from the model conceptualization process. State-transition (Markov) models are well suited to modeling transitions between multiple chronic comorbidities and recurrent acute events. The key limitation of Markov models is that they are memory-less, meaning patient attributes and disease history do not influence transitions between health states. This limitation can be overcome through individual-based modeling, in which a single individual's trajectory in the model is tracked from birth until death and individual attributes and disease history are incorporated in prediction equations for transition probabilities. We will develop an individual-based state transition model with a one-year time interval for model cycles, as more precise timing of events is unlikely to be influential over a lifetime horizon.

In assessing the face validity of a schematic of the proposed model structure, experts highlighted the central role of pain in SCD. Although vaso-occlusive episodes (VOE) were

**Table 2. Lifetime prevalence of disease attributes among individuals diagnosed with SCD in Medicaid, Medicare, and dual-eligibility cohorts.**

| | Medicaid (N = 39,366) | Medicare (6,522) | Dual-eligible (N = 36,846) | Combined (N = 82,734) |
|---|---|---|---|---|
| *Acute Events* | | | | |
| Vaso-occlusive pain episodes | 78.1% | 42.4% | 71.8% | 72.5% |
| Stroke | 9.1% | 21.4% | 15.4% | 12.9% |
| Fever | 61.4% | 29.5% | 55.6% | 56.3% |
| Splenic disease | 11.7% | 5.8% | 8.6% | 9.8% |
| Priapism | 3.5% | 1.0% | 3.9% | 3.5% |
| Dactylitis | 2.9% | 5.0% | 5.1% | 4.1% |
| Acute chest syndrome | 31.6% | 15.6% | 32.0% | 30.5% |
| Myocardial infarction | 0.9% | 5.7% | 3.8% | 2.5% |
| Infections | 79.4% | 73.0% | 82.1% | 80.1% |
| Acute renal failure | 7.9% | 30.9% | 22.4% | 16.2% |
| Multi-organ failure | 2.0% | 5.6% | 6.0% | 4.0% |
| Bacteremia and sepsis | 17.5% | 20.1% | 28.4% | 22.6% |
| Acute anemia | 8.5% | 5.4% | 10.4% | 9.1% |
| *Subacute Disorders* | | | | |
| Chronic pain | 18.0% | 31.0% | 39.6% | 28.7% |
| Fatigue | 20.2% | 57.4% | 44.7% | 34.0% |
| Asthma | 36.5% | 24.3% | 37.1% | 35.8% |
| Leg ulcers | 0.9% | 3.9% | 3.1% | 2.1% |
| Hepatobilliary complications and liver disease | 19.2% | 22.6% | 25.5% | 22.3% |
| Sleep disordered breathing and nocturnal hypoxemia | 25.8% | 29.6% | 32.5% | 29.1% |
| Depression and unspecified psychosis | 15.1% | 28.3% | 32.3% | 23.8% |
| *Chronic Disorders* | | | | |
| Chronic renal disease | 14.2% | 54.0% | 35.1% | 26.6% |
| Pulmonary hypertension and cardiovascular diseases | 35.5% | 71.1% | 59.6% | 49.0% |
| Chronic lung diseases | 21.8% | 38.3% | 36.3% | 29.6% |
| Ocular complications | 4.2% | 9.7% | 7.3% | 6.0% |
| Cognitive impairment | 9.3% | 15.7% | 12.1% | 11.1% |
| Avascular necrosis & bone damage | 11.9% | 14.9% | 20.3% | 15.9% |

included as an acute event in the final list of disease attributes, it is only one dimension of pain [52]. Chronic pain, which includes neuropathic pain and pain of other etiologies, can vary in intensity and frequency over time but is a near universal feature of SCD that significantly impacts health-related quality of life (HrQoL), medical and non-medical resource use [52,53]. To reflect this, we will model chronic pain separately from other chronic disorders. Individuals can experience chronic pain throughout the model time horizon. Because pain episodes result in healthcare encounters in a minority of cases, and to avoid underestimating the prevalence of chronic pain and VOEs using claims data, we will conduct external validation with published estimates and calibrate the prediction equations through curve fitting if necessary [54].

### 3.4. The MEASURE model

A schematic of the conceptual model structure is presented in **Fig 2**. In this model, a 'health state' is defined by the presence or absence of a condition or event. Unlike a typical Markov model, in which a patient can exist in any one of mutually exclusive health states, patients in the model can experience several different health states at a given point in time. This is necessary to avoid the combinatorial explosion of modeling 26 mutually exclusive disease attributes. Instead, we defined four dimensions of disease: 1) chronic pain, 2) acute events, 3) chronic

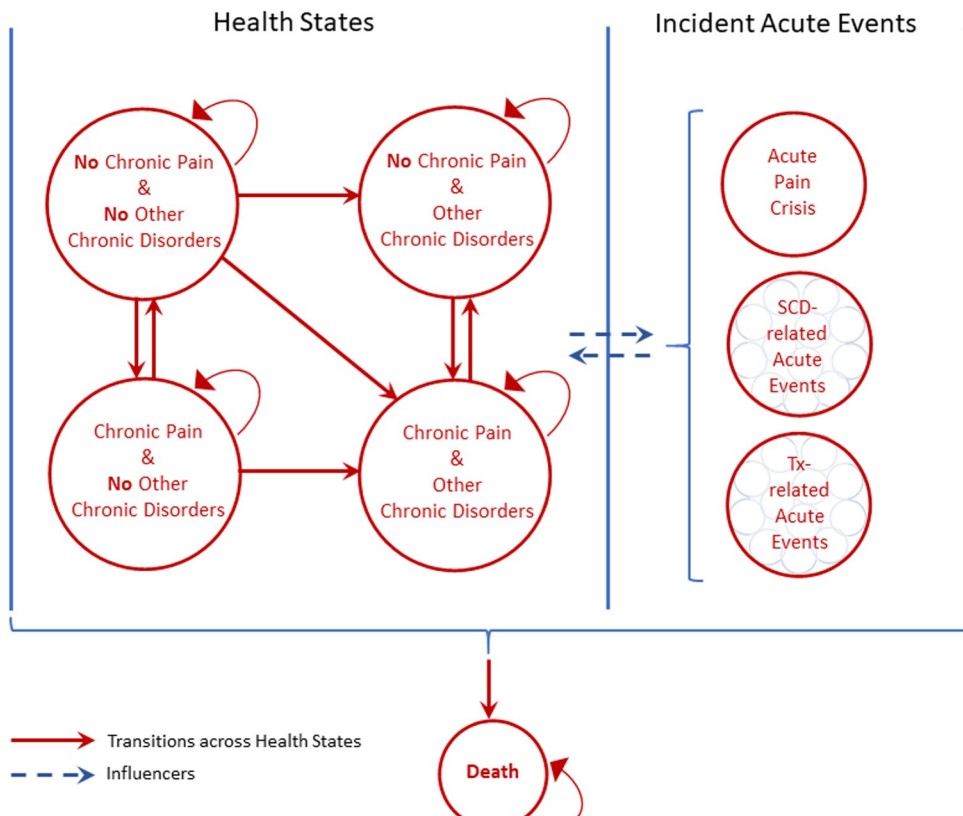

**Fig 2. Diagram of the conceptual model of SCD.** Treatment (Tx)- related acute events are leukopenia, thrombocytopenia, and oligospermia/azospermia associated with hydroxyurea; iron overload, transfusion reaction and infection associated with transfusion; graft versus host disease, graft failure, bronchiolitis obliterans, osteoporosis, iron overload, depression, posterior reversible encephalopathy syndrome, post-transplant lymphoproliferative disorder, and secondary malignancy associated with HSCT; and graft failure, bronchiolitis obliterans, osteoporosis, iron overload, depression, posterior reversible encephalopathy syndrome, post-transplant lymphoproliferative disorder, and secondary malignancy associated with genetic therapies.

disorders, and 4) treatment complications. Each dimension excepting chronic pain consists of multiple health states. The specific health states within the chronic disorders and acute events dimension were determined through model conceptualization. Treatment complications are common adverse events associated with hydroxyurea and transfusion, part of Common Care for SCD [55], and comparator curative therapies, HSCT, and gene therapies, determined from clinical guidelines [55,56]. (The complete set of disease attributes and treatment complications included in each dimension are shown in Table A1 in S1 Appendix.) The presence or absence of chronic pain occurs in parallel with other health states. A patient's health at any point in time is determined by the combination of all health states across all four dimensions, which includes a health state for no major complications. Individuals can transition or progress between health states over one-year model cycles. Because acute events are short in duration, we model them as instantaneous events, meaning transitions and progression do not occur. Mortality can occur in any health state.

### 3.5. Parameter estimation and calibration

To overcome the simplistic state-dependency of traditional Markov models, age- and sex-specific transitions between health states, treatment use, treatment complications, and healthcare

utilization will be determined using prediction indices. The first set of indices will predict the incidence of each acute event and chronic disorder as a function of patient health status in the previous year (chronic pain, chronic disorders, and acute events), disease history (history of chronic pain and acute events, duration of chronic disorders), treatment use (hydroxyurea, acute or chronic transfusion) and treatment complications in the previous year. The second set of indices will predict treatment use based on health status in the current year, disease history, treatment use and treatment complications in the previous year. Similarly, treatment complications will be predicted based on health status and treatment uses in the current year, disease history, and treatment complications in the previous year. Finally, health utilization will be a function of health status, disease history, treatment use, and treatment complications in the current year. Patient demographics will include sex, age-specific indicators, and birth-cohort indicators. A graphical depiction of the relations between variables in the prediction indices is shown in Figure A2 in S1 Appendix. We will apply a regularized regression approach (e.g., LASSO or elastic net) to manage the parameter space. We will split the sample into a training set (50%), and two test sets for out-of-sample calibration of the prediction indices (25%), and the final decision model (25%). Parameter uncertainty will be based on out-of-sample bootstrap replicates in the first test set.

We will use the private and public payer claims datasets described in Section 2.6 to inform the prediction indices. To reflect differences in the characteristics of publicly and privately insured individuals, which could preclude generalizations between these populations, we will generate separate prediction equations for each database. This will also help us overcome limitations in the sociodemographic data available for enrollees in claims data. Instead, patient attributes will be implicitly represented based on established differences between payer populations [57].

## 3.6. Treatment effects

Genetic therapies are an emerging technology with uncertainty around their effectiveness in broad patient populations, the durability of clinical benefits, and the potential for treatment-related complications. In a landscape analysis conducted by members of the CEIA team, 14 clinical trials of 10 gene therapies for SCD were identified. Case reports were available for 8 patients [58]. We will apply this preliminary evidence on the proportion of patients who were symptom free and the proportion experiencing treatment-related complications as risk ratios in the decision model. Following the eligibility criteria of several clinical trials, we will initially restrict treatment to patients with severe SCD, defined based on their history of severe adverse events [58]. The cost of treatment will be based on the price of genetic therapy for beta-thalassemia [14,59]. We will use scenario analyses to account for considerable uncertainty in the price of genetic therapies, its effectiveness over a patient's lifetime and among patients with less severe SCD, and the potential for induced mutagenesis and other treatment-related complications. For HSCT, the annual costs, effectiveness, and treatment-related complications will be directly assessed in the private and public payer claims datasets previously described, and uncertainty will be evaluated using probabilistic sensitivity analysis.

## 4. Discussion

We report on the development of a conceptual model that captures major chronic conditions and acute events associated with SCD over a patient's lifetime as determined from roundtable discussions with experts. Our conceptual model contains a more comprehensive set of disease attributes and treatment complications than has been included in previous cost-effectiveness models of SCD. In an analysis of Medicare, Medicaid, and dual-eligibility claims databases, there were no disease attributes with extremely low prevalence. Experts recommended an

individual-based state-transition model as the most appropriate modeling technique. We identified additional data sources and literature estimates to validate the prevalence of chronic and acute pain observed in claims databases, obtain disease-specific data on health-related quality of life, and capture non-medical costs for patients and caregivers.

The conceptual model developed here will be used to assess the cost-effectiveness of newer non-curative therapies, genetic therapies, and HSCT versus Common Care for SCD. By using an individual-based model, we can account for variation in health outcomes and resource use and assess the distribution of value among subgroups of patients. We will parameterize the conceptual model with patterns of treatment and age and sex-specific trajectories of disease burden using separate claims databases for individuals with Medicare, Medicaid, dual-eligibility, and commercial insurance. This will allow us to replicate cost-effectiveness analyses between payer populations, reflecting differences in patient characteristics and potential mechanisms for reimbursement. By taking the societal perspective for our analysis, we will consider the impact of curative therapies on a wide range of resources, including time-use costs and the indirect economic burden of productivity loss for patients and caregivers.

Previous cost-effectiveness models of SCD were not developed through an explicit model conceptualization process. This may have contributed to the small number of disease attributes identified, which were limited to specific treatment complications, typically related to blood transfusion, and none or a very narrow set of comorbidities [31]. In addition to assessing face validity by disease experts, we confirmed cross validity of our model structure by ensuring it contained all chronic conditions and acute events in previous cost-effectiveness models. Because gene therapies can affect multiple aspects of disease over a patient's lifetime, excluding potentially relevant disease attributes could lead to inaccurate estimates of their value.

We have developed the first conceptual model for evaluating specific curative therapies for SCD. A previous modeling study assessed the value of a hypothetical curative therapy for patients with SCD that is administered at birth and completely eliminates all disease-related complications [51]. The authors developed a Markov model with separate health states for mild, moderate, and severe SCD based on the frequency of VOE. Additional disease attributes were not considered, and their analysis was conducted from a U.S. payer perspective, meaning the indirect costs of SCD were excluded. Including the impact of a perfectly curative therapy on a wider range of costs and health effects is likely to improve its value. In contrast, accounting for treatment complications in the model structure and the potential for partial disease remission would decrease its favorability.

## 4.1 Limitations

Previous model conceptualization studies have conducted elicitation exercises to define patient attributes that are most influential for disease progression [22,24]. We did not conduct this second component of model conceptualization. The claims databases we established as primary model inputs have limited sociodemographic information on enrollees available, and as a result, only age, sex, and region of residence can be included in prediction indices for health state transitions, adverse events, treatment patterns, and healthcare utilization. This is in line with previous studies that recommend simplifying the conceptual model when data are not available [22–24]. Although explicitly modeling all potentially relevant patient attributes could improve our estimation of trajectories of disease burden, we will overcome this limitation by generating separate parameter sets for public and private payer databases, which will implicitly account for differences in these populations and capture the majority of individuals with prevalent SCD in the U.S. Despite using data on the population of publicly insured individuals with SCD in the U.S., we face a sparsity of data in specific subgroups of patients characterized

by their comorbidities. Employing machine learning techniques can overcome some of these limitations by identifying parsimonious models that are still highly predictive of outcomes, and predictions will be calibrated through split sample techniques.

Using the population of publicly insured individuals with SCD in the U.S. will help us accurately model health outcomes and conduct population-level and specific sub-population level evaluations. The explicit conceptualization process, as presented here, helps make the black-box nature of simulation models more transparent to policymakers. However, our model may not be well suited to inform individual-level clinical care, unless it is calibrated in the future against individual-level clinical trial data.

Although we will take a societal perspective for our cost-effectiveness analysis, as per guidelines from the Second Panel on CEA, our analyses will not be able to address all aspects of the sociodemographic and institutional burdens faced due to SCD. We will also not be able to inform advancements in clinical care for these patients directly. However, we will generate evidence on the value of using certain therapies for specific sub-groups of patients at certain time points during their disease trajectory, which we hope will inform the advancement of clinical care and insurance coverage of appropriate treatments.

We did not elicit expert opinion using structured methods like the Delphi technique, which has been employed in previous model conceptualization studies [22,24]. However, roundtable discussions can generate a wider range of opinions without the need to establish consensus. Given the wide scope of curative therapies, discussion groups are better suited to capturing all potentially relevant disease attributes. To avoid the potential for a few experts to dominate the discussion and prevent all opinions from being heard, we formed several smaller groups and imposed a time limit for discussions.

Finally, there are inherent limitations to using data on the short-term effectiveness of gene therapies found in clinical trials to project their impact on disease burden over a lifetime horizon. However, our work is part of a movement towards incorporating decision modeling earlier in clinical research so that barriers to practice implementation and sustainability can be addressed [60]. Our conceptual model is intended as a framework for exploring uncertainties in the emerging effectiveness data. It is expected to be continuously updated using the best available evidence.

## 5. Conclusions

This study established a conceptual model for assessing the cost-effectiveness of curative therapies for SCD. Experts identified a wide range of chronic conditions and acute events to include in the model structure based on their impact on the quality-adjusted life years and resource use of patients. Our conceptual model had face validity in expert assessment and parallel validity with other cost-effectiveness models of SCD. By conducting an explicit process of model conceptualization, we aimed to incorporate input from a wide range of stakeholders, clearly outline the model development process, and increase the transparency of model results.

## Supporting information

**S1 Appendix.**
(DOCX)

## Acknowledgments

We acknowledge feedback from the National Heart, Lung and Blood Institute, participants from Emmes, the CureSC Expert Panel and the CureSC Initiative. The Cure Sickle Cell

Initiative (CureSCi) is a collaborative, patient-focused research effort designed to accelerate the advancement of genetic-based cures for sickle cell disease. The Initiative is funded by the National Heart, Lung, and Blood Institute (NHLBI), part of the National Institutes of Health (NIH). We are grateful to Joshua Roth for his input on earlier versions of this conceptualization. All errors and opinions are ours.

## Author Contributions

**Conceptualization:** Kate M. Johnson, Boshen Jiao, M. A. Bender, Scott D. Ramsey, Beth Devine, Anirban Basu.

**Formal analysis:** Kate M. Johnson, Boshen Jiao.

**Methodology:** Kate M. Johnson, Anirban Basu.

**Supervision:** M. A. Bender, Scott D. Ramsey, Anirban Basu.

**Validation:** Kate M. Johnson, Boshen Jiao, M. A. Bender, Scott D. Ramsey, Beth Devine, Anirban Basu.

**Writing – original draft:** Kate M. Johnson.

**Writing – review & editing:** Kate M. Johnson, Boshen Jiao, M. A. Bender, Scott D. Ramsey, Beth Devine, Anirban Basu.

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
