## [Decision Letter · Decision Letter 0]

10 Mar 2022

PONE-D-22-01351Development of a Conceptual Model for Evaluating Curative Therapies for Sickle Cell DiseasePLOS ONE

Dear Dr. Basu,

Thank you for submitting your manuscript to PLOS ONE. After careful consideration, we feel that it has merit but does not fully meet PLOS ONE’s publication criteria as it currently stands. Therefore, we invite you to submit a revised version of the manuscript that addresses the points raised during the review process.

We look forward to receiving your revised manuscript.

Kind regards,

Mohamed A Yassin, MD

Academic Editor

PLOS ONE

Journal Requirements:

 “YES. National Heart, Lung and Blood Institute, Cure Sickle Cell initiative. This research was funded in part by the National Institutes of Health (NIH) Agreements OT3HL152448 and OT3HL151434. The views and conclusions contained in this document are those of the authors and should not be interpreted as representing the official policies, either expressed or implied, of the NIH.”

“The Initiative is funded by the National Heart, Lung, and Blood Institute (NHLBI), part of the National Institutes of Health (NIH). We are grateful to Joshua Roth for his input on earlier versions of this conceptualization.”

“YES. National Heart, Lung and Blood Institute, Cure Sickle Cell initiative. This research was funded in part by the National Institutes of Health (NIH) Agreements OT3HL152448 and OT3HL151434. The views and conclusions contained in this document are those of the authors and should not be interpreted as representing the official policies, either expressed or implied, of the NIH.”

Additional Editor Comments:

Major revision required.

Reviewers' comments:

Reviewer's Responses to Questions

**Comments to the Author**

1. Does the manuscript provide a valid rationale for the proposed study, with clearly identified and justified research questions?

Reviewer #1: Yes

Reviewer #2: Yes

2. Is the protocol technically sound and planned in a manner that will lead to a meaningful outcome and allow testing the stated hypotheses?

Reviewer #1: No

Reviewer #2: Yes

3. Is the methodology feasible and described in sufficient detail to allow the work to be replicable?

Reviewer #1: No

Reviewer #2: Yes

4. Have the authors described where all data underlying the findings will be made available when the study is complete?

Reviewer #1: No

Reviewer #2: Yes

5. Is the manuscript presented in an intelligible fashion and written in standard English?

Reviewer #1: Yes

Reviewer #2: Yes

6. Review Comments to the Author

You may also provide optional suggestions and comments to authors that they might find helpful in planning their study.

Reviewer #1: In this article, Anirban Baru and colleagues developed a conceptual model for the evaluation of curative therapies in Sickle Cell Disease (SC). The model was built and discussed using consultations with the patients and health care providers and experts and also by conducting intensive literature review. The model included four chronic events associated with SCD: chronic pain, acute events, chronic conditions and treatment complications. In general, having this model will be helpful in future analysis of SCD-targeting curative therapies. The model’s description is vague and hard to grasp and needs better description. Also, it seems to miss recent therapies that are being incorporated into SCD treatment.

Major critiques:

1. Please dedicate a section specifically for the description of the model.

2. A visual presentation of the literature review would be helpful.

3. While blood transfusion and hydroxyurea were considered, no others emerging therapies were included such as voxelotor or crizanlizumab treatments.

Minor changes

1. References are not in the PLoS One format.

Reviewer #2: The cited work appears well researched, rationale and reproducible. Outcome measures were generally justifiable for addressing the questions posed by the authors.

Patient data retrieval from online databases is an attractive option in view of the project’s mandate. Likewise, the multidisciplinary team proposed should prove beneficial in navigating the nuances of sickle cell disease prognoses and economic/societal impact and feasibility.

A few areas of note arose; these were generally recognized by the authors and highlighted under ‘limitations’ of the study.

Firstly, although the prediction analyses appear robust, the depth of the data (age, gender and region) intended for the prediction of healthcare utilization, health state transitions, adverse events and the likelihood of effective treatment appear somewhat tenuous for the useful simulation of health state fluxes in SCD. The etiology of the disease with its diversity of physiological manifestations should pose unique challenges for the reliable imputation of clinical trajectories. Furthermore, the immunological depression further veils the true impact of SCD-related events, giving rise to above-normal incidences of SCD-independent comorbidities.

Secondly, the appropriateness of roundtable meetings as supplants to consensus from experienced personnel is ambiguous. Again, consequent to the convoluted nature and frequency of the acute and chronic clinical symptomology of SCD, a more prudent ‘real-world’ approach might strengthen the efficacy of a conceptual predictive model.

Thirdly, the value of the ‘societal’ definition remarked on by the authors is uncertain. The conceptual framework outlined, whilst likely agreeable with the cost-effectiveness ascertainment facet of the work, appears less geared to optimizing the quality of care and clinical outcomes of the target population. The societal view may be skewed away from the advancement of clinical care and patient outcomes towards the tangential sociodemographic and institutional burdens imposed by the patients’ `disease.

Together with the narrowness of the accessible patient data, omission of traditional decision hierarchy, the overall societal benefit of this explicit conceptualization model is not apparent. I support the work and recognize its potential benefit with some minor revisions.

7. PLOS authors have the option to publish the peer review history of their article (what does this mean?). If published, this will include your full peer review and any attached files.

Reviewer #1: No

Reviewer #2: **Yes: **Andre S.S. Bowers

---

## [Author Response · Author response to Decision Letter 0]

29 Mar 2022

Please see attached document titled "Response to Reviewers"

---

## [Editor Report · Decision Letter 1]

11 Apr 2022

Development of a Conceptual Model for Evaluating New Non-Curative and Curative Therapies for Sickle Cell Disease

PONE-D-22-01351R1

Dear Dr. Basu,

We’re pleased to inform you that your manuscript has been judged scientifically suitable for publication and will be formally accepted for publication once it meets all outstanding technical requirements.

Kind regards,

Mohamed A Yassin, MD

Academic Editor

PLOS ONE

Additional Editor Comments (optional):

The revision is satisfactory to proceed with publication
---

## [Editor Report · Acceptance letter]

20 Apr 2022

PONE-D-22-01351R1 

Development of a Conceptual Model for Evaluating New Non-Curative and Curative Therapies for Sickle Cell Disease 

Dear Dr. Basu:

I'm pleased to inform you that your manuscript has been deemed suitable for publication in PLOS ONE. Congratulations! Your manuscript is now with our production department. 

Kind regards, 

on behalf of

Dr. Mohamed A Yassin 

Academic Editor

PLOS ONE